# DMark: Order-Agnostic Watermarking for Diffusion Large Language Models

## Abstract

Diffusion large language models (dLLMs) offer faster generation than autoregressive models while maintaining comparable quality, but existing watermarking methods fail on them due to their non-sequential decoding. Unlike autoregressive models that generate tokens left-to-right, dLLMs can finalize tokens in arbitrary order, breaking the causal design underlying traditional watermarks. We present DMark, the first watermarking framework designed specifically for dLLMs. DMark introduces three complementary strategies to restore watermark detectability: predictive watermarking uses model-predicted tokens when actual context is unavailable; bidirectional watermarking exploits both forward and backward dependencies unique to diffusion decoding; and predictive-bidirectional watermarking combines both approaches to maximize detection strength. Experiments across multiple dLLMs show that DMark achieves $92.0 - 99.5\%$ detection rates at $1\%$ false positive rate while maintaining text quality, compared to only $49.6 - 71.2\%$ for naive adaptations of existing methods. DMark also demonstrates robustness against text manipulations, establishing that effective watermarking is feasible for non-autoregressive language models.

## 1 Introduction

Large language models (LLMs) (OpenAI et al., 2024; Comanici et al., 2025) have become indispensable infrastructure across education, media, and software development, fundamentally reshaping how we create and interact with text. While autoregressive (AR) LLMs currently dominate the landscape, diffusion-based LLMs (dLLMs) have emerged as a compelling alternative, offering more than $10\times$ faster inference speed while maintaining comparable generation quality (Inception Labs, 2025). This new paradigm has gained significant traction, with commercial systems like Mercury Coder (Inception Labs, 2025), Gemini Diffusion (Google Deepmind, 2025), and Seed Diffusion (Song et al., 2025) demonstrating production-ready capabilities, alongside open-source implementations including LLaDA (Nie et al., 2025), LLaDA 1.5 (Zhu et al., 2025), and DREAM (Ye et al., 2025).

As dLLMs rapidly gain adoption, establishing *text provenance* mechanisms becomes critical for detecting AI-generated content, deterring plagiarism, and ensuring responsible disclosure (Liu et al., 2024; Zhao et al., 2025). Watermarking, which embeds statistically detectable signals in generated text, has proven effective for traditional autoregressive (AR) LLMs (Qu et al., 2025; Zhao et al., 2023). However, these methods fail catastrophically on dLLMs due to their fundamentally different generation process.

The core challenge lies in how existing methods assume sequential generation. KGW (Kirchenbauer et al., 2023), the most widely adopted watermarking approach, uses preceding tokens to determine how to watermark the current token. This works for AR models but breaks in dLLMs, which generate through iterative denoising: starting from fully masked sequences, they compute logits for all positions simultaneously and update tokens in arbitrary order. Positions can be filled out-of-sequence and refined across multiple steps, violating the sequential dependency and stable prefix assumptions that AR watermarks require.

To address this fundamental incompatibility, we present DMark, the first watermarking framework designed for dLLMs, built on two key observations about their generation process. First, since

dLLMs compute logits for all positions simultaneously, we can predict missing context tokens directly from their logit distributions, even when actual tokens are unavailable. Second, since dLLMs finalize tokens in arbitrary order rather than left-to-right, we can exploit bidirectional dependencies: not only can preceding tokens determine watermarking, but subsequent tokens can also constrain their predecessors.

These observations directly motivate three watermarking strategies: **Predictive watermarking** leverages parallel logit computation to infer missing context, ensuring watermark injection even without neighboring tokens; **Bidirectional watermarking** exploits arbitrary generation order by using both forward green lists (based on preceding tokens) and backward green lists (based on subsequent tokens); **Predictive-bidirectional watermarking** combines both strategies, predicting unavailable context while applying bidirectional constraints to maximize watermark signal strength.

In summary, our contributions are three-fold:

- **First watermarking formalization for dLLMs.** We formalize watermarking for diffusion language models and demonstrate why existing AR methods fail catastrophically, achieving only $49.6 - 71.2\%$ detection rates at $1\%$ FPR due to out-of-order generation and iterative refinement that violate sequential dependencies.

- **Novel bidirectional and predictive watermarking methods.** We introduce three strategies exploiting dLLM properties: predictive watermarking leveraging parallel logits to infer missing context, bidirectional watermarking using both forward and backward dependencies unique to dLLMs, and their synergistic combination achieving $92.0 - 99.5\%$ detection rates while preserving generation quality.

- **Comprehensive evaluation across models and attacks.** We evaluate DMark on multiple dLLMs and datasets, demonstrating robustness against text manipulations while establishing optimal parameter configurations for different security-quality trade-offs.

## 2 PRELIMINARIES

### 2.1 DIFFUSION LARGE LANGUAGE MODELS (DLLMS)

Unlike autoregressive models that generate tokens sequentially as $p(x_i|x_{<i})$, dLLMs generate text through iterative denoising over $T$ steps. Starting from a fully masked sequence $\mathbf{x}^{(T)} = [\text{MASK}]^n$, the model progressively refines tokens:

$$\mathbf{x}^{(t-1)} \sim p_\theta(\mathbf{x}^{(t-1)}|\mathbf{x}^{(t)}), \quad t = T, T-1, \ldots, 1 \tag{1}$$

At each step $t$, the model computes logits for all positions simultaneously:

$$\mathbf{L}^{(t)} = f_\theta(\mathbf{x}^{(t)}) \tag{2}$$

Crucially, positions can be updated in arbitrary order based on confidence scores $c_i^{(t)} = \max_v L_{i,v}^{(t)}$, or any other remasking strategies, enabling parallel generation (Nie et al., 2025; Ye et al., 2025).

In the low-confidence remasking strategy, the generation process generally involves: (1) selecting positions to unmask based on confidence, (2) sampling tokens for selected positions, and (3) potentially remasking low-confidence tokens for refinement. This out-of-order generation means position $i$ may be filled before position $i - 1$, and any token can be overwritten across multiple steps, fundamentally breaking the sequential assumptions of AR watermarking.

### 2.2 WATERMARKING FOR AUTOREGRESSIVE MODELS

To understand why existing watermarking fails on dLLMs, we examine the KGW method (Kirchenbauer et al., 2023), the most widely adopted watermarking approach for AR LLMs.

Given vocabulary $\mathcal{V}$, KGW partitions tokens into *green* and *red* lists based on preceding context. KGW embeds the watermark signal into generated text by increasing the generation likelihood of several pseudo-randomly chosen tokens. When generating the token at the $i$-th position, KGW uses

a hash function $h$ seeded with key $s$ and preceding context $\mathbf{x}_{<i} = (x_{i-w}, \ldots, x_{i-1})$ where $w \geq 1$ is the context window size, to partition the vocabulary $\mathcal{V}$ into a *green* token list $\mathcal{G}_i$ and a *red* token list $\mathcal{R}_i$:

$$h_i = h(s, \mathbf{x}_{<i})$$
$$\mathcal{G}_i = \{v \in \mathcal{V} : p(h_i, v) < \gamma\}, \quad \mathcal{R}_i = \mathcal{V} \setminus \mathcal{G}_i \tag{3}$$

where $\gamma \in (0, 1)$ controls the green list ratio, and $p$ is a pseudo-random function that maps each token $v$ to $[0, 1)$. During generation, KGW biases the logits by adding $\delta > 0$ to green tokens:

$$\tilde{L}_{i,v} = \begin{cases} L_{i,v} + \delta & \text{if } v \in \mathcal{G}_i \\ L_{i,v} & \text{if } v \in \mathcal{R}_i \end{cases} \tag{4}$$

In watermark detection, it computes a z-score based on the proportion of green tokens:

$$z = \frac{|\{i : x_i \in \mathcal{G}_i\}| - \gamma n}{\sqrt{\gamma(1 - \gamma)n}} \tag{5}$$

This method crucially depends on sequential generation where preceding context $\mathbf{x}_{<i}$ is always available when generating $x_i$, which is an assumption violated in dLLMs. The algorithm for KGW watermarking is detailed in Appendix A.1.

## 3 METHODS

We develop four watermarking methods with increasing sophistication for dLLMs' non-sequential generation. **Direct adaptation** (§3.1) naively applies KGW when preceding context exists, achieving limited watermark signals as out-of-order generation leaves many positions unwatermarked. **Predictive watermarking** (§3.2) leverages parallel logit computation to predict missing context, enabling watermarking at all positions despite prediction errors. **Bidirectional watermarking** (§3.3) shifts from sequential to bidirectional paradigm, exploiting both forward and backward dependencies to generate green lists in both directions. **Predictive-bidirectional watermarking** (§3.4) synergizes prediction with bidirectional constraints, maximizing detection strength across all generation orders.

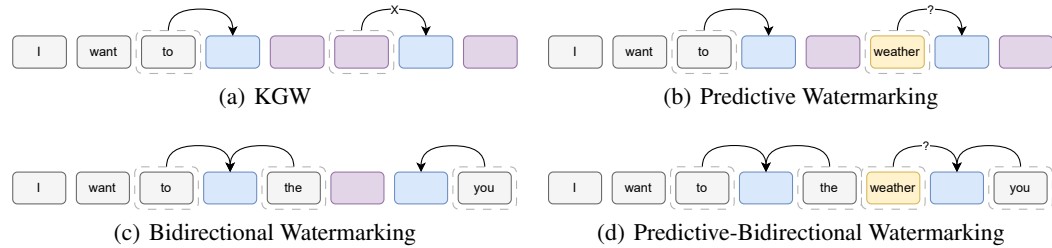

Figure 1: Illustration of four watermarking methods for dLLMs. Gray rectangles represent finalized tokens, purple rectangles represent unfinalized tokens, yellow rectangles represent unfinalized but predicted tokens, and blue rectangles represent tokens to be generated. (a) KGW watermarking applies watermarks only when preceding context exists. (b) Predictive watermarking uses predicted preceding tokens as context when actual tokens are unavailable. (c) Bidirectional watermarking leverages both forward green lists and backward green lists. (d) Predictive-bidirectional watermarking combines prediction with bidirectional green lists for maximum watermark signals.

### 3.1 DIRECT ADAPTATION OF KGW

We first consider a straightforward adaptation of KGW to dLLMs, which simply applies KGW conditionally: when preceding context $x_{i-1}$ exists, we watermark using the green list $\mathcal{G}_i$; otherwise, we generate without watermarking. We focus on single-token context ($x_{i-1}$ only) as longer contexts are rarely available in dLLMs' out-of-order generation. In Appendix A.2 we detail this approach.

This approach suffers from context availability: watermarks can only be applied when preceding context $x_{i-1}$ exists. Since dLLMs generate positions out of order, position $i$ often lacks its predecessor $x_{i-1}$, preventing watermark injection. Our experiments with LLaDA (Nie et al., 2025) on ELI5 (Fan et al., 2019) confirm this limitation, where only 67% of tokens had available preceding context, resulting in weak watermark signals.

### 3.2 PREDICTIVE WATERMARKING

Our key insight for overcoming missing context is to leverage dLLMs' unique parallel logit computation. Unlike AR models, dLLMs compute logits $\mathbf{L}^{(t)}$ for all positions simultaneously at each denoising step, including unfilled positions. We propose to exploit this property by predicting missing context tokens directly from their logit distributions:

$$\hat{x}_{i-1}^{(t)} = \arg\max_v L_{i-1,v}^{(t)} \tag{6}$$

We then construct the green list $\mathcal{G}_i$ using this predicted token $\hat{x}_{i-1}^{(t)}$, enabling watermark injection even when actual context is unavailable. While incorrect predictions yield weaker watermark signals, accurate predictions enable proper watermark embedding. This novel strategy ensures watermark injection at every position regardless of generation order, substantially improving upon direct adaptation. The complete algorithm is detailed in Appendix A.3.

### 3.3 BIDIRECTIONAL WATERMARKING

While predictive watermarking ensures watermarking signals will be embedded regardless of generation order, its effectiveness is limited by prediction accuracy. During diffusion, logit distributions shift substantially as context solidifies. For example, with noisy context "[MASK] [MASK] network", the foremost token might initially predict "the" (a common pattern), but as denoising reveals "deep [MASK] network", the actual token becomes "neural", causing us to watermark using the wrong green list $\mathcal{G}_i(\text{the})$ instead of $\mathcal{G}_i(\text{deep})$.

To address this drawback, we take a different approach: instead of relying solely on forward context, we exploit dLLMs' unique bidirectional conditioning capability. We begin by examining the traditional **forward detection objective**:

$$\max \sum_{i \in [n]} \mathbf{1}[x_i \in \mathcal{G}_i] \tag{7}$$

where $\mathcal{G}_i = \{v \in \mathcal{V} : p(h(x_{i-1}, s), v) \leq \gamma\}$.

This forward-only approach uses preceding context $x_{i-1}$ to generate the green list $\mathcal{G}_i$, indicating that watermark detectability depends solely on prior tokens. While natural for autoregressive models, this constraint is unnecessarily restrictive for dLLMs, which can condition on tokens in both forward and backward direction. The key insight is that dLLMs' bidirectional nature enables a complementary backward watermarking process: instead of asking that whether $x_i$ is in the green list of $x_{i-1}$, we can equally ask that whether $x_i$ is in a set that makes $x_{i+1}$ green-listed.

Formally, according to the definition of green list, which will be referred to as **forward green list** $\mathcal{G}_i$, we define the **backward green list** $\mathcal{G}_i'$ as the set of tokens at position $i$ that would cause the subsequent token $x_{i+1}$ to be in its forward green list:

$$\mathcal{G}_i' = \{v \in \mathcal{V} : x_{i+1} \in \mathcal{G}_{i+1}(v)\} \tag{8}$$

$$\mathcal{R}_i' = \{v \in \mathcal{V} : x_{i+1} \in \mathcal{R}_{i+1}(v)\} \tag{9}$$

The backward green list enables an equivalent detection objective, which will be referred to as **backward detection objective**:

$$\max \sum_{i \in [n]} \mathbf{1}[x_i \in \mathcal{G}_i'] \tag{10}$$

This dual perspective ensures effective watermarking across all generation scenarios: we apply forward constraints when $x_{i-1}$ exists, backward constraints when $x_{i+1}$ exists, or both when surrounded

by context, adapting dynamically to available neighbors. We formalize this bidirectional watermarking as follows:

$$\tilde{z}_{i,v} = z_{i,v} + \delta \cdot B(v, x_{i-1}^{(t)}, x_{i+1}^{(t)}), \tag{11}$$

where the bias term $B(v, x_{i-1}^{(t)}, x_{i+1}^{(t)})$ is defined as:

$$B(v, x_{i-1}^{(t)}, x_{i+1}^{(t)}) = \begin{cases} \mathbf{1}[v \in \mathcal{G}_i] & \text{if } x_{i-1}^{(t)} \text{ exists, } x_{i+1}^{(t)} \text{ does not exist} \\ \mathbf{1}[v \in \mathcal{G}_i] + \mathbf{1}[v \in \mathcal{G}_i'] & \text{if both } x_{i-1}^{(t)} \text{ and } x_{i+1}^{(t)} \text{ exist} \\ \mathbf{1}[v \in \mathcal{G}_i'] & \text{if } x_{i-1}^{(t)} \text{ does not exist, } x_{i+1}^{(t)} \text{ exists} \\ 0 & \text{if neither } x_{i-1}^{(t)} \text{ nor } x_{i+1}^{(t)} \text{ exists} \end{cases} \tag{12}$$

This formulation adapts to any generation order: forward bias for left-to-right, backward bias for right-to-left, and bidirectional bias when both neighbors exist, accommodating dLLMs' non-sequential generation patterns. The algorithm is detailed in Appendix A.4.

### 3.4 PREDICTIVE-BIDIRECTIONAL WATERMARKING

The bidirectional and predictive strategies are orthogonal and can be naturally combined. While bidirectional watermarking exploits both forward and backward context, predictive watermarking uses predicted tokens when actual ones are unavailable. Predictive-bidirectional watermarking applies both techniques simultaneously: it uses predictions for missing neighbors, applying bidirectional constraints by using both forward and backward green lists. This combination maximizes watermark coverage across all generation scenarios, achieving the highest detection rates by leveraging every available signal source.

Formally, we define the forward and backward green lists with prediction as:

$$\hat{\mathcal{G}}_i = \begin{cases} \{v \in \mathcal{V} : p(h(s, x_{i-1}^{(t)}), v) \leq \gamma\} & \text{if } x_{i-1}^{(t)} \text{ exists} \\ \{v \in \mathcal{V} : p(h(s, \hat{x}_{i-1}), v) \leq \gamma\} & \text{otherwise, where } \hat{x}_{i-1} = \arg\max_v z_{i-1,v} \end{cases} \tag{13}$$

$$\hat{\mathcal{G}}_i' = \begin{cases} \{v \in \mathcal{V} : p(h(s, v), x_{i+1}^{(t)}) \leq \gamma\} & \text{if } x_{i+1}^{(t)} \text{ exists} \\ \{v \in \mathcal{V} : p(h(s, v), \hat{x}_{i+1}) \leq \gamma\} & \text{otherwise, where } \hat{x}_{i+1} = \arg\max_v z_{i+1,v} \end{cases} \tag{14}$$

The watermark bias for predictive-bidirectional is formalized as:

$$\tilde{z}_{i,v} = z_{i,v} + \delta \cdot B_{\text{pred}}(v, x_{i-1}^{(t)}, x_{i+1}^{(t)}) \tag{15}$$

where the predictive-bidirectional bias term is:

$$B_{\text{pred}}(v, x_{i-1}^{(t)}, x_{i+1}^{(t)}) = \mathbf{1}[v \in \hat{\mathcal{G}}_i] + \mathbf{1}[v \in \hat{\mathcal{G}}_i'] \tag{16}$$

Note that unlike pure bidirectional watermarking which may have no bias when neighbors are absent, predictive-bidirectional always applies bias by using predicted tokens to construct both $\hat{\mathcal{G}}_i$ and $\hat{\mathcal{G}}_i'$, ensuring watermark injection at every position. We demonstrate it in Algorithm 1.

## 4 EXPERIMENTS

### 4.1 SETUPS

**Models and Datasets**  We evaluate DMark on three open-source dLLMs: **LLaDA-Instruct-8B** (Nie et al., 2025), **LLaDA-1.5-8B** (Zhu et al., 2025), and **Dream-v0-Instruct-7B** (Ye et al., 2025). Following previous watermarking studies (Zhao et al., 2023; Pan et al., 2024), we use two complementary benchmarks: **ELI5** (Fan et al., 2019) for question-answering tasks and **C4** (Raffel et al., 2023) for text completion with 30-token prefixes. We generate 500 samples per dataset, filtering for sequences with at least 200 tokens as in (Kirchenbauer et al., 2023) and excluding instances with abnormal repetition patterns in baseline.

---

**Algorithm 1** Predictive-Bidirectional Watermarking for dLLMs

---

**Require:** Token sequence $\mathbf{x}^{(t)}$, position $i$, secret seed $s$, green ratio $\gamma$, bias strength $\delta$
**Ensure:** Watermarked token $x_i^{(t+1)}$
1: **if** $x_{i-1}^{(t)}$ exists **then**                                         ▷ Determine forward green list
2:     $\mathcal{G}_i \leftarrow \{v \in \mathcal{V} : p(h(s, x_{i-1}^{(t)}), v) \leq \gamma\}$
3: **else**
4:     $\hat{x}_{i-1} \leftarrow \arg\max_v \text{Model}(\mathbf{x}^{(t)}, i-1)_v$
5:     $\mathcal{G}_i \leftarrow \{v \in \mathcal{V} : p(h(s, \hat{x}_{i-1}), v) \leq \gamma\}$
6: **if** $x_{i+1}^{(t)}$ exists **then**                                       ▷ Determine backward green list
7:     $\mathcal{G}_i' \leftarrow \{v \in \mathcal{V} : p(h(s, v), x_{i+1}^{(t)}) \leq \gamma\}$
8: **else if** $i < n$ **then**
9:     $\hat{x}_{i+1} \leftarrow \arg\max_v \text{Model}(\mathbf{x}^{(t)}, i+1)_v$
10:    $\mathcal{G}_i' \leftarrow \{v \in \mathcal{V} : p(h(s, v), \hat{x}_{i+1}) \leq \gamma\}$
11: **else**
12:    $\mathcal{G}_i' \leftarrow \mathcal{V}$                                            ▷ No constraint at sequence end
13: $\mathbf{z}_i \leftarrow \text{Model}(\mathbf{x}^{(t)}, i)$
14: $\tilde{z}_{i,v} \leftarrow z_{i,v} + \delta \cdot (\mathbf{1}[v \in \mathcal{G}_i] + \mathbf{1}[v \in \mathcal{G}_i'])$ for all $v \in \mathcal{V}$
15: **return** $x_i^{(t+1)} \sim \text{Softmax}(\tilde{\mathbf{z}}_i)$

---

**Implementation** For LLaDA inference, we use 256 denoising steps with 32-token blocks to generate 256-token sequences, with temperature set to $0.0$ for deterministic evaluation. For Dream inference, we also use 256 denoising steps to generate 256-token sequences, with all other hyperparameters set to the default values. It's worth noting that Dream doesn't support block generation. To enable efficient bidirectional watermarking, we precompute a bit matrix $\mathcal{M} \in \{0,1\}^{|\mathcal{V}| \times |\mathcal{V}|}$ encoding green list relationships. Row $i$ stores the forward green list of token $i$ (where $\mathcal{M}_{ij} = 1$ if token $j$ is green given $i$), while column $j$ stores the backward green list for token $j$ (where $\mathcal{M}_{ij} = 1$ if token $i$ makes $j$ green). This dual encoding enables $O(1)$ green list retrieval via simple row/column lookups, avoiding costly vocabulary iterations during generation. Detection uses the z-score from Equation 5 with a predefined threshold. Texts that exceed this threshold are classified as watermarked. To ensure fair comparison, all baseline methods and our method use identical hyperparameters and detection thresholds computed from the non-watermarked instances.

### 4.2 Watermark Methods Performance

**Detection Effectiveness Across Methods.** We evaluate detection effectiveness using true positive rate (TPR) at three critical false positive rate (FPR) thresholds: $0.5\%$, $1\%$, and $5\%$. These low FPR values ensure minimal false accusations against human-written text, which is essential for practical deployment where incorrectly flagging legitimate content poses serious concerns. We compare four watermarking methods (KGW, Predictive, Bidirectional, and Predictive Bidirectional) across three dLLMs and two datasets to assess performance under diverse generation conditions.

Table 1 reveals that traditional watermarking fails on diffusion models: KGW achieves only $49.6-71.2\%$ TPR at $1\%$ FPR across settings, while Predictive marginally improves to $54.2-79.3\%$. Bidirectional methods outperform unidirectional approaches, reaching $74.2 - 88.0\%$ TPR. Meanwhile, Predictive-Bidirectional achieves near-perfect 92.0-99.5% TPR, demonstrating that leveraging both forward and backward context is essential for reliable watermark detection in non-sequential generation.

**Impact of Generation Length.** Watermark detection fundamentally relies on statistical accumulation of biased token choices, making generation length a critical factor for real-world viability. Short texts like social media posts offer limited statistical evidence, while longer documents like articles provide more detection opportunities—yet both scenarios demand reliable watermarking. We empirically examine how detection strength scales with text length to establish minimum length requirements.

Table 1: Detection performance comparison of watermarking methods across different dLLMs. Experiments with $n = 200$ tokens, $\gamma = 0.5$, $\delta = 2.0$, and low confidence remasking method.

| Method | C4 TPR (%) | | | ELI5 TPR (%) | | |
|---|---|---|---|---|---|---|
| | @0.5% FPR | @1% FPR | @5% FPR | @0.5% FPR | @1% FPR | @5% FPR |
| *LLaDA-1.5-8B* | | | | | | |
| KGW | 30.4 | 71.2 | 88.0 | 45.8 | 63.6 | 88.4 |
| Predictive | 36.4 | 76.8 | 91.0 | 60.2 | 77.6 | 94.2 |
| Bidirectional | 53.4 | 87.6 | 95.4 | 79.2 | 86.2 | 97.2 |
| Predictive Bidirectional | **80.2** | **96.8** | **98.2** | **95.8** | **99.0** | **99.8** |
| *LLaDA-Instruct-8B* | | | | | | |
| KGW | 29.6 | 49.6 | 83.6 | 54.4 | 65.0 | 84.4 |
| Predictive | 36.4 | 54.2 | 89.2 | 63.8 | 76.0 | 92.6 |
| Bidirectional | 55.0 | 74.2 | 95.2 | 78.4 | 88.0 | 95.8 |
| Predictive Bidirectional | **80.0** | **92.0** | **98.4** | **97.8** | **99.2** | **99.8** |
| *Dream-v0-Instruct-7B* | | | | | | |
| KGW | 32.8 | 54.4 | 75.9 | 41.7 | 59.5 | 83.5 |
| Predictive | 45.2 | 70.3 | 89.3 | 63.2 | 79.3 | 92.1 |
| Bidirectional | 67.6 | 82.0 | 93.6 | 73.3 | 85.6 | 95.5 |
| Predictive Bidirectional | **93.6** | **97.2** | **98.1** | **98.8** | **99.5** | **100.0** |

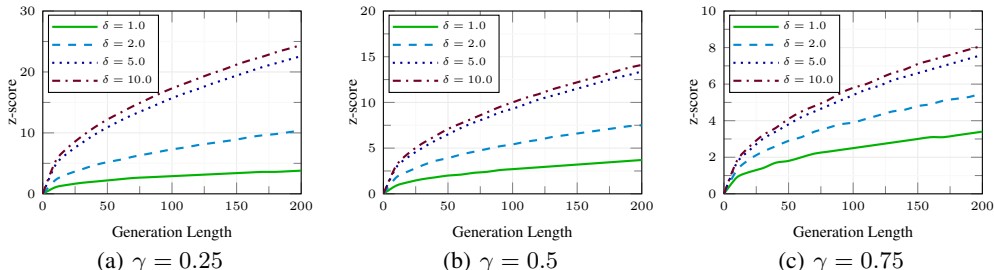

(a) $\gamma = 0.25$      (b) $\gamma = 0.5$      (c) $\gamma = 0.75$

Figure 2: Impact of generation length on watermark detection strength (z-score) for different watermark parameters. Results shown for Predictive-Bidirectional watermarking on LLaDA-Instruct-8B with ELI5 dataset and different green list ratios ($\gamma$). Higher z-scores indicate stronger watermark detection, while lower PPL indicates better text quality.

Figure 2 demonstrates how watermark detection strength scales with text length. Short texts ($L < 50$) require stronger watermark parameters ($\delta \geq 5.0$) to achieve reliable detection due to limited statistical evidence. Medium-length texts ($50 \leq L \leq 150$) achieve practical detection with moderate settings, making them suitable for typical applications. Long texts ($L > 150$) enable robust detection even with conservative parameters, achieving z-scores exceeding 20 under standard configurations.

**Trade-off between Watermark Effectiveness and Text Quality.** We then investigate how watermark detectability scales while maintaining text quality. We use Llama-3-8B-Instruct as our reference model for calculating perplexity (PPL). As demonstrated in Figure 3, low watermark strength with $\delta \leq 2.0$ succeeds to achieve a balance between watermark detectability and text quality, which is sufficient for low FPR detection while maintaining PPL. While high watermark strength $\delta \geq 5.0$ achieves perfect detection, it significantly degrades text quality, which is not practical for real-world deployment.

### 4.3 WATERMARK ROBUSTNESS

Real-world deployment faces adversarial threats ranging from benign text corruptions to sophisticated paraphrasing attacks. We evaluate DMark's resilience across this threat spectrum to understand its security boundaries and guide parameter selection for adversarial environments.

Table 2 evaluates watermarking methods' resilience against adversarial attacks. Predictive-Bidirectional consistently outperforms all baselines, achieving $95 - 97\%$ TPR at $1\%$ FPR against

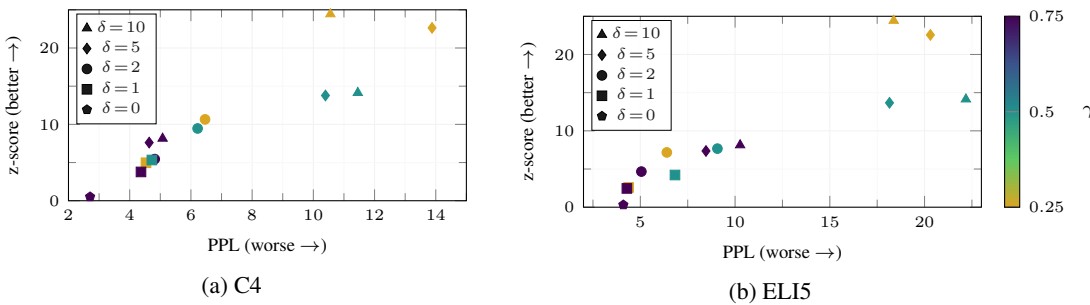

Figure 3: Quality-detectability trade-off for varying watermark configurations. Each point represents a different combination of bias strength $\delta$ and green list ratio $\gamma$.

Table 2: Robustness evaluation of watermarking strategies against various attacks. Values show TPR (%) at different FPR thresholds. We detail the paraphrasing attack setup in Appendix B.

| Attack Type | Param. | TPR @ 0.5% FPR | | | | TPR @ 1% FPR | | | | TPR @ 5% FPR | | | |
|---|---|---|---|---|---|---|---|---|---|---|---|---|---|
| | | KGW | Predict | Bidir | PBidir | KGW | Predict | Bidir | PBidir | KGW | Predict | Bidir | PBidir |
| *Delete* | 10% | 36.8 | 40.0 | 63.2 | **93.8** | 51.2 | 52.2 | 76.8 | **97.2** | 74.4 | 81.0 | 91.2 | **99.4** |
| | 20% | 21.0 | 25.2 | 45.8 | **78.8** | 35.4 | 39.2 | 62.2 | **88.4** | 68.4 | 68.0 | 84.6 | **98.2** |
| *Insert* | 10% | 34.0 | 37.0 | 60.4 | **91.8** | 46.0 | 52.0 | 73.8 | **95.6** | 73.2 | 76.2 | 90.0 | **98.8** |
| | 20% | 22.0 | 25.6 | 41.0 | **79.4** | 34.4 | 35.4 | 54.8 | **86.8** | 59.4 | 64.0 | 82.0 | **96.6** |
| *Swap* | 10% | 35.2 | 40.6 | 57.0 | **91.0** | 49.0 | 53.6 | 69.2 | **96.8** | 69.4 | 76.6 | 91.0 | **99.4** |
| | 20% | 19.4 | 24.0 | 34.8 | **75.4** | 30.8 | 37.4 | 47.8 | **84.2** | 59.8 | 64.4 | 76.8 | **95.8** |
| *Substitution* | 10% | 31.8 | 37.8 | 57.2 | **91.2** | 45.0 | 52.2 | 69.6 | **95.0** | 71.4 | 77.4 | 88.4 | **99.4** |
| | 20% | 15.2 | 20.6 | 31.2 | **73.0** | 26.6 | 31.8 | 47.8 | **83.6** | 51.8 | 62.0 | 78.0 | **95.0** |
| *Paraphrase* | GPT | 9.8 | 12.8 | 19.8 | **41.0** | 17.0 | 21.0 | 29.6 | **51.2** | 44.2 | 46.8 | 62.0 | **80.2** |
| | Dipper | 20.2 | 23.2 | 34.6 | **61.6** | 29.6 | 32.4 | 45.8 | **72.6** | 54.4 | 60.0 | 70.2 | **89.2** |

10% token-level attacks, compared to only $45 - 53\%$ for sequential baselines (KGW, Predictive). Even under aggressive 20% token corruption, Predictive-Bidirectional maintains $83 - 88\%$ TPR while KGW drops to $26 - 35\%$. Vulnerability to paraphrasing attacks is a well-known limitation across LLM watermarking methods (Zhang et al., 2024). Dipper (Krishna et al., 2023) paraphrasing at 20% diversity reduces Predictive-Bidirectional to $72.6\%$ TPR, while GPT-5-nano paraphrasing yields only $51.2\%$ TPR for our best approach.

### 4.4 PARAMETER SENSITIVITY

Practical deployment requires understanding how green list ratio $\gamma$ and bias strength $\delta$ interact to balance detection reliability with generation quality. We systematically evaluate these parameters to identify optimal configurations for different use cases. Table 3 reveals critical trade-offs between detection effectiveness and text quality. Weak watermarking with $\delta = 1.0$ maintains excellent text quality with PPL below $4.5$ but fails to provide reliable detection across all green list ratios. Moderate strength at $\delta = 2.0$ shows that smaller green lists perform better: $\gamma = 0.25$ achieves $96.6\%$ TPR on C4 while $\gamma = 0.75$ drops to just $60.6\%$. Text quality remains acceptable at this strength level, with PPL staying below $6.5$ across all configurations. Strong watermarking with $\delta \geq 5.0$ guarantees near-perfect detection but at substantial quality cost—PPL exceeds 20 on ELI5 with $\gamma = 0.25$. Based on these results, we recommend $\gamma = 0.5$ with $\delta = 2.0$ for practical deployment, achieving over $92\%$ detection rates while preserving readable text quality.

## 5 RELATED WORK

### 5.1 DLLMS

Diffusion Large Language Models (dLLMs) (Yu et al., 2025) extend the diffusion framework (Ho et al., 2020; Nichol & Dhariwal, 2021; Song et al., 2020) from traditional image and video generation (Podell et al., 2023) to natural language. Unlike autoregressive models that decode sequentially,

Table 3: Parameter sensitivity analysis of DMark watermarking system. All experiments use Predictive-Bidirectional watermarking on LLaDA-8B-Instruct with $n = 200$ tokens.

| $\gamma$ | $\delta$ | C4 TPR (%) | | | C4 | ELI5 TPR (%) | | | ELI5 |
|---|---|---|---|---|---|---|---|---|---|
| | | @0.5% FPR | @1% FPR | @5% FPR | PPL | @0.5% FPR | @1% FPR | @5% FPR | PPL |
| 0.25 | 1.0 | 13.4 | 31.4 | 61.6 | 2.89 | 22.2 | 37.2 | 64.2 | 4.38 |
| | 2.0 | 89.8 | 96.6 | 98.4 | 4.04 | 96.8 | 98.2 | 99.8 | 6.40 |
| | 5.0 | 100.0 | 100.0 | 100.0 | 10.90 | 100.0 | 100.0 | 100.0 | 20.31 |
| | 10.0 | 100.0 | 100.0 | 100.0 | 10.56 | 100.0 | 100.0 | 100.0 | 18.37 |
| 0.5 | 1.0 | 13.0 | 27.6 | 67.4 | 2.94 | 30.4 | 41.8 | 66.0 | 4.47 |
| | 2.0 | 80.0 | 92.0 | 98.4 | 3.98 | 97.8 | 99.2 | 99.8 | 6.34 |
| | 5.0 | 100.0 | 100.0 | 100.0 | 7.56 | 100.0 | 100.0 | 100.0 | 15.15 |
| | 10.0 | 100.0 | 100.0 | 100.0 | 6.86 | 100.0 | 100.0 | 100.0 | 13.30 |
| 0.75 | 1.0 | 7.6 | 12.2 | 54.2 | 2.92 | 10.6 | 27.8 | 50.4 | 4.29 |
| | 2.0 | 41.4 | 60.6 | 94.8 | 3.41 | 83.0 | 93.2 | 98.0 | 5.05 |
| | 5.0 | 98.8 | 99.6 | 99.8 | 4.64 | 100.0 | 100.0 | 100.0 | 8.46 |
| | 10.0 | 100.0 | 100.0 | 100.0 | 5.08 | 100.0 | 100.0 | 100.0 | 10.27 |

dLLMs generate text through iterative denoising, where a noisy sequence is progressively reconstructed. Early studies on discrete diffusion, including D3PM (Austin et al., 2021), RDM (Zheng et al., 2023), DiffusionBERT (Austin et al., 2021), MD4 (Shi et al., 2024) and MDLM (Sahoo et al., 2024), explored different objectives, noise schedules, and parameterizations, largely at the billion-parameter scale. These works established the feasibility of applying diffusion to text and multi-modal tasks, setting the stage for larger systems. Recent progress has focused on scaling dLLMs, with models that rival or even outperform autoregressive LLMs while often delivering faster inference. Representative advances include LLaDA (Nie et al., 2025), the first large-scale DLLM, and DIFFUSION-LLMs (Ye et al., 2023) with multi-stage training. DiffuGPT/DiffuLLaMA (Gong et al., 2024) adapt pretrained autoregressive models into the diffusion paradigm, and DREAM (Ye et al., 2025) further underscores DLLMs' capability in reasoning-intensive tasks. More recent developments, such as LLaDA 1.5 (Zhu et al., 2025) with variance-reduced preference optimization and TESS 2 (Tae et al., 2025) with autoregressive initialization and adaptive noise scheduling, continue to enhance both efficiency and generation quality.

## 5.2 WATERMARKING FOR LLMS

Most text watermarking for LLMs follows the green/red-list method of Kirchenbauer et al. (2023): a secret key slightly upweights a "green" subset during sampling, and detection checks for an over-representation of green tokens. Follow-up work improves the quality–power trade-off with variance reduction (Hu et al., 2023), adapts the bias to model uncertainty (Liu & Bu, 2024), and provides finite-sample guarantees (Zhao et al., 2023), while other variants aim to stay hard for third-party detectors yet verifiable by the key holder (Christ et al., 2024). SynthID-Text (Dathathri et al., 2024) shows a production deployment at scale with calibrated thresholds and measured quality impact. Attacks reveal practical limits: reverse engineering can recover keys or green lists (Jovanović et al., 2024), and scrubbing can remove the signal while preserving utility (Chen et al., 2025).

## 6 CONCLUSION

We presented DMark, the first watermarking framework tailored for diffusion language models, addressing the urgent need for dLLM watermarking. By recognizing that dLLMs' parallel generation pattern enables bidirectional context exploitation and token predictions, we fundamentally shift watermarking from sequential dependencies to predictive bidirectional forward-backward constraints. Our Predictive-Bidirectional method achieves $92.0 - 99.5\%$ detection rates at $1\%$ FPR, substantially outperforming traditional approaches ($49.6 - 71.2\%$). Meanwhile, our framework shows strong resilience against common text manipulations. This work not only provides practical tools for watermarking dLLM-generated text but also establishes theoretical foundations for watermarking non-sequential generative models, paving the way for responsible deployment of next-generation text synthesis systems.

ETHICS STATEMENT

This work does not involve human subjects, personal data, or sensitive information. All datasets used in our experiments (C4 and ELI5) are publicly available benchmark datasets. We strictly adhered to ethical research practices and did not conduct any data collection that could raise privacy, security, or fairness concerns. Our work focuses on a new watermarking framework for dLLMs, without introducing risks of harmful applications. To the best of our knowledge, this research complies with the ICLR Code of Ethics and poses no foreseeable ethical concerns.

REPRODUCIBILITY STATEMENT

We have made extensive efforts to ensure the reproducibility of our work. Comprehensive implementation details are reported in Section 4.1 and the detailed algorithm for watermarking is provided in Section 3 and Appendix A. Upon acceptance, we will release the code of our method to facilitate replication and further research.

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

APPENDIX

# A   ADDITIONAL ALGORITHMS FOR WATERMARKING

## A.1   KGW WATERMARKING FOR AR MODELS

---

**Algorithm 2** KGW Watermarking

---

**Require:** Context $\mathbf{x}_{<i} = (x_{i-w}, \ldots, x_{i-1})$, secret seed $s$, green ratio $\gamma$, bias $\delta$
**Ensure:** Watermarked token $x_i$
  1: Compute context hash: $h_i \leftarrow h(s, \mathbf{x}_{<i})$
  2: Partition vocabulary: $\mathcal{G}_i \leftarrow \{v \in \mathcal{V} : p(h_i, v) \leq \gamma\}$
  3: Get model logits: $\mathbf{z}_i \leftarrow \mathrm{Model}(\mathbf{x}_{<i})$
  4: Apply bias: $\tilde{z}_{i,v} \leftarrow z_{i,v} + \delta \cdot \mathbf{1}[v \in \mathcal{G}_i]$ for all $v \in \mathcal{V}$
  5: Sample: $x_i \sim \mathrm{Softmax}(\tilde{\mathbf{z}}_i)$
  6: **return** $x_i$

---

## A.2   KGW WATERMARKING FOR dLLMs

---

**Algorithm 3** Direct Adaptation of KGW for dLLMs

---

**Require:** Token sequence $\mathbf{x}^{(t)}$ at timestep $t$, position $i$, secret seed $s$, green ratio $\gamma$, bias strength $\delta$
**Ensure:** Watermarked token $x_i^{(t+1)}$
  1: **if** position $i - 1$ has been generated **then**
  2:    Compute context hash: $h_i \leftarrow h(s, x_{i-1}^{(t)})$
  3:    Partition vocabulary: $\mathcal{G}_i \leftarrow \{v \in \mathcal{V} : p(h_i, v) \leq \gamma\}$
  4:    Get model logits: $\mathbf{z}_i \leftarrow \mathrm{Model}(\mathbf{x}^{(t)}, i)$
  5:    Apply watermark bias: $\tilde{z}_{i,v} \leftarrow z_{i,v} + \delta \cdot \mathbf{1}[v \in \mathcal{G}_i]$ for all $v \in \mathcal{V}$
  6:    Sample token: $x_i^{(t+1)} \sim \mathrm{Softmax}(\tilde{\mathbf{z}}_i)$
  7: **else** *// No adjacent context available, generate without watermark*
  8:    Get model logits: $\mathbf{z}_i \leftarrow \mathrm{Model}(\mathbf{x}^{(t)}, i)$
  9:    Sample token: $x_i^{(t+1)} \sim \mathrm{Softmax}(\mathbf{z}_i)$
 10: **return** $x_i^{(t+1)}$

---

## A.3   PREDICTIVE WATERMARKING

---

**Algorithm 4** Predictive Watermarking for dLLMs

---

**Require:** Token sequence $\mathbf{x}^{(t)}$ at timestep $t$, position $i$, secret seed $s$, green ratio $\gamma$, bias strength $\delta$
**Ensure:** Watermarked token $x_i^{(t-1)}$
  1: **if** position $i - 1$ has been generated **then**
  2:    $\hat{x}_{i-1}^{(t)} \leftarrow x_{i-1}^{(t)}$
  3: **else** *// Predict token using logits*
  4:    Get logits for position $i - 1$: $\mathbf{z}_{i-1} \leftarrow \mathrm{Model}(\mathbf{x}^{(t)}, i-1)$
  5:    Predict most likely token: $\hat{x}_{i-1} \leftarrow \arg\max_{v \in \mathcal{V}} z_{i-1,v}$
  6: Partition vocabulary: $\mathcal{G}_i \leftarrow \{v \in \mathcal{V} : p(h(s, \hat{x}_{i-1}), v) \leq \gamma\}$
  7: Get model logits for position $i$: $\mathbf{z}_i \leftarrow \mathrm{Model}(\mathbf{x}^{(t)}, i)$
  8: Apply watermark bias: $\tilde{z}_{i,v} \leftarrow z_{i,v} + \delta \cdot \mathbf{1}[v \in \mathcal{G}_i]$ for all $v \in \mathcal{V}$
  9: Sample watermarked token: $x_i^{(t-1)} \sim \mathrm{Softmax}(\tilde{\mathbf{z}}_i)$
 10: **return** $x_i^{(t-1)}$

---

## A.4 BIDIRECTIONAL WATERMARKING

---

**Algorithm 5** Bidirectional Watermarking for dLLMs

---

**Require:** Token sequence $\mathbf{x}^{(t)}$ at timestep $t$, position $i$, secret key $s$, green ratio $\gamma$, bias strength $\delta$
**Ensure:** Watermarked token $x_i^{(t+1)}$
 1: Initialize: $\mathcal{G}_i \leftarrow \mathcal{V}, \mathcal{G}_i' \leftarrow \mathcal{V}$
 2: **if** $x_{i-1}^{(t)}$ exists **then**
 3:     $\mathcal{G}_i \leftarrow \{v \in \mathcal{V} : p(h(s, x_{i-1}^{(t)}), v) \leq \gamma\}$
 4: **if** $x_{i+1}^{(t)}$ exists **then**
 5:     $\mathcal{G}_i' \leftarrow \{v \in \mathcal{V} : p(h(s, v), x_{i+1}^{(t)}) \leq \gamma\}$
 6: Get model logits: $\mathbf{z}_i \leftarrow \text{Model}(\mathbf{x}^{(t)}, i)$
 7: Apply bias: $\tilde{z}_{i,v} \leftarrow z_{i,v} + \delta \cdot \mathbf{1}[v \in \mathcal{G}_i] + \delta \cdot \mathbf{1}[v \in \mathcal{G}_i']$ for all $v \in \mathcal{V}$
 8: Sample: $x_i^{(t-1)} \sim \text{Softmax}(\tilde{\mathbf{z}}_i)$
 9: **return** $x_i^{(t-1)}$

---

## B PARAPHRASING ATTACK SETUP

To evaluate the robustness of our watermarking method against paraphrasing attacks, we employ state-of-the-art language models, GPT-5-nano, to rewrite watermarked text while preserving semantic content. The following prompt is used for all paraphrasing experiments:

```
Please paraphrase the following text while preserving its
meaning.  Output only the rewritten text, nothing else:

[WATERMARKED TEXT]
```

Meanwhile, when using Dipper model to paraphrase, we use lexical diversity 20%, order diversity 0%, and sentence interval 3 to simulate the real world paraphrasing scenario.

## C WATERMARK GENERATION EXAMPLES

This section presents concrete examples demonstrating DMark's impact on text generation quality and watermark detection strength. Table 4 compares outputs from identical prompts with and without watermarking, illustrating how our method maintains semantic coherence while embedding detectable signals. The examples span both C4 text continuation and ELI5 question-answering tasks, showcasing DMark's effectiveness across different generation contexts.

Each example includes ground truth text for reference, along with z-scores quantifying watermark strength and perplexity measuring text quality. Non-watermarked texts exhibit near-zero z-scores as expected, while watermarked versions achieve z-scores between $6.37$ and $9.33$, well above typical detection thresholds. Despite this strong watermark signal, perplexity increases modestly from $2.32 - 3.75$ to $3.43 - 6.01$, confirming that watermarking preserves readable, coherent text while enabling reliable detection.

## D LLM USAGE

In this section, we clarify the role of large language models (LLMs) in preparing this work. The model was used exclusively for language polishing, such as refining grammar, style, and readability, without contributing to the research design, analysis, or conclusions.

Table 4: Examples of watermarked and non-watermarked text generation from DMark. First two examples come from C4 dataset, and the last two examples come from ELI5 dataset. Contents are truncated for readability.

| Prompt | Ground Truth | Non-Watermarked | Watermarked | Non-WM | | WM | |
|---|---|---|---|---|---|---|---|
| | | | | z-score | PPL | z-score | PPL |
| *Students are shouting several slogans to guarantee the school's autonomy in front of Jogyesa Temple. On April 15th, 2016,* | the General Student Council and Student Council of Post-graduate held the 4.15 Jogye rally. The rally was also held in the last year with the same objectives. [...truncated] | the students of Jogyesa High School held a protest in front of Jogyesa Temple in support of the school's autonomy. The students shouted several slogans to express their demands, including School's autonomy is our right and School's autonomy is our future. [...truncated] | the students and the teachers of Jogyesa High School gathered in front of Jogyesa Temple to express their demand for their school to be independent. They demanded that the government separates their school from the Jogyesa High School Management Committee. [...truncated] | 0.00 | 3.26 | 9.33 | 4.08 |
| *Locksmith Westington – Putting in place the best of locking and keying services is just one part of the story. However the challenge lies in keeping* | the locking and keying systems in good condition at all points of time. Without proper maintenance it is quite obvious that the locks and keys would become damaged and replacing the same with new ones could be a big drain on resources and it would also take lot of time. [...truncated] | your property safe and secure. That's why we offer a range of security services to ensure your property is protected at all times. At Locksmith Westington, we understand the importance of protecting your property from theft and damage. That's why we offer a range of security services to ensure your property is protected at all times. [...truncated] | up to date with the latest trends and technology that are affecting the world of locks and keys. At locksmith Westington, it has always been a priority to ensure that the locks and keys we provide are up to the latest standards of quality and performance. [...truncated] | -2.26 | 2.32 | 7.50 | 3.43 |
| *Why does metal react so violently when microwaved?* | The way microwaves work is through jiggling charged/polar particles in your food (the water primarily). This jiggling increases their temperature and that heats up the rest of your food. Thats why you cant heat oil as easily as you can water. However, metals like iron are great conductors of electrons. [...truncated] | Metal reacts violently when microwaved because it is an excellent conductor of heat. When microwaves are applied to a food, they cause the molecules in the food to vibrate rapidly, generating heat. However, when microwaves are applied to a metal, the metal absorbs the microwaves and heats up quickly. The rapid heating of the metal can cause it to melt, spark, or even catch fire. [...truncated] | Metal does not react violently when microwaved, but it does absorb heat and can burn or catch on fire. When microwaves are emitted by the microwave, they vibrate and cause molecules and atoms to vibrate and absorb energy, increasing heat. The primary reason why metal can cause problems when microwaved is that it is a good conductor of electricity and heat, which means it can absorb and transfer heat quickly. [...truncated] | 0.56 | 3.54 | 7.07 | 6.01 |
| *Why is it when oil prices go up gas prices immediately go up but when oil prices come down the price of gas never comes down as fast as when the price of oil increases?* | On TV, when the price of gas goes up, they say they have no choice but to sell it higher in gas stations too. But when the price goes down, they say that they had already bought a lot of gas when it was higher so they cannot lower the prices immediately or theyd lose money on it. There is probably a good explanation for it but my guess is that its another of these Heads I win, Tails, you lose situation ;) | When oil prices go up, gas prices immediately go up because gas prices are closely tied to oil prices. Gasoline is typically derived from crude oil, and the cost of producing gasoline is directly related to the cost of crude oil. When oil prices rise, the cost of producing gasoline increases, which in turn drives up the price of gasoline. However, when oil prices come down, the price of gas never comes down as fast as when the price of oil increases. [...truncated] | When oil prices go up, they are considered to be an indicator of an improving economy. This leads to an increase in demand for oil. Gasoline, being a major component of oil, also sees increased demand. The increased demand for gasoline causes the price of gas to rise, creating a direct correlation between oil and gas prices. On the other hand, when oil prices come down, they are seen as a sign of an improving economy. [...truncated] | 0.0 | 3.75 | 6.37 | 5.67 |