# OpenReview forum: "DMark: Order-Agnostic Watermarking for Diffusion Large Language Models"
_ICLR.cc/2026/Conference — ICLR 2026 Conference Withdrawn Submission_

### Official Review · Reviewer_cMT7 · 2025-10-15

**Soundness:** 1
**Presentation:** 3
**Contribution:** 2
**Rating:** 2
**Confidence:** 5

**Summary:**

This paper proposes a watermarking framework for diffusion large language models (dLLMs)  and introduces three strategies—Predictive Watermarking, Bidirectional Watermarking, and Predictive-Bidirectional Watermarking, to improve the detectability of the watermark. The effectiveness of these strategies is validated through experiments on multiple dLLMs and datasets.

**Strengths:**

1. The paper is written clearly and exhibits good logical flow.
2. The paper attempts to design watermarking solutions specifically for dLLMs, taking into account the characteristics of bidirectional context.

**Weaknesses:**

1. The paper lacks sufficient baseline comparisons, such as the relevant Unigram algorithm and the PatternMark algorithm, which is designed for order-agnostic language models.
2. The experiments in this paper are insufficient, lacking evaluation on downstream tasks and comparisons with watermarking performance in autoregressive LLMs.

**Questions:**

1. There are two errors in the description of the dLLM generation process (lines 96-100). First, dLLMs generate tokens for all positions at each timestep and then perform remasking, rather than generating tokens only for unmasked positions. Second, not every token can be masked at each timestep; positions that were unmasked in previous timesteps will not be considered again, otherwise the generation process would not terminate.
2. Regarding lines 172-173, it should be noted that dLLMs inherently generate tokens for all positions at each timestep, which is not an original concept proposed in this paper. For reference, please refer to Figure 2c in the paper "Large Language Diffusion Models".
3. Among the three strategies proposed in the paper, it seems that only the Predictive Bidirectional strategy is effective, while the other two strategies perform poorly. Why does the combination of the two strategies lead to a significant improvement in performance?
4. A key issue is not addressed in the paper: How is watermark detection performed? Since the detection process cannot restore the context from the generation, how is the vocabulary partition determined?

---

### Official Review · Reviewer_pbXE · 2025-10-21

**Soundness:** 2
**Presentation:** 3
**Contribution:** 2
**Rating:** 4
**Confidence:** 4

**Summary:**

This paper proposes DMark, a watermarking framework specifically designed for diffusion large language models (dLLMs), which generate text through iterative denoising rather than sequential left-to-right generation. DMark introduces three complementary strategies: predictive watermarking, bidirectional watermarking, and their combination.

**Strengths:**

- First systematic watermarking solution for diffusion language models.

**Weaknesses:**

- The core contribution is essentially adapting token-level watermarking to accommodate dLLMs' generation characteristics rather than proposing fundamentally new watermarking principles.

- The paper shows perplexity increases (from 4.5 to 6.3 on ELI5 with δ=2.0), which undermines a key advantage of dLLMs. If watermarking negates the quality benefits that justify using dLLMs over autoregressive models, practitioners may prefer well-established autoregressive LLMs with mature KGW watermarking instead.

- Table 2 reveals catastrophic failure against semantic-preserving rewrites: only 51.2% TPR against GPT paraphrasing and 72.6% against Dipper at 1% FPR, compared to 99.2% on clean text. This is a fundamental limitation because adversaries can trivially evade detection by paraphrasing through widely available LLMs, rendering the watermark ineffective for the stated goals of "detecting AI-generated content".

- The paper does not report generation speed, memory footprint, or latency comparisons between watermarked and non-watermarked dLLMs.

**Questions:**

- Could you provide empirical analysis of prediction accuracy at different generation stages (early vs. late denoising steps) and its quantitative impact on z-scores?

- Have you explored potential defenses such as semantic watermarking, embedding signals in latent representations, or using error-correcting codes to recover from partial watermark corruption?

- Could you provide formal analysis showing that forward and backward green lists provide independent or complementary detection signals?

- Could you provide head-to-head comparisons of "dLLM+DMark" versus "AR-LLM+KGW" on the same tasks, measuring detection rate, text quality (perplexity), and generation speed?

---

### Official Review · Reviewer_VNpz · 2025-10-24

**Soundness:** 3
**Presentation:** 2
**Contribution:** 2
**Rating:** 2
**Confidence:** 3

**Summary:**

This paper investigates the problem of instantiating watermarks for diffusion language models as opposed to autoregressive models. A watermark is a statistical signal hidden inside text that can be detected by anyone with access to a secret key but is intended to not distort the quality of text, thus remaining undetectable to observers without access to a secret key. Many schemes have been instantiated for autoregressive models by modifying the sampling procedure of models, with a prominent such scheme being the green list approach, where a hash function looks at the recent context and returns a pseudorandom subset of the vocabulary to upweight in generation. While this has proven effective in autoregressive models, it is not immediate how to instantiate this scheme with masked diffusion models.  This paper suggests two approaches to solving this problem.  First, it suggests considering a bidirectional hashing scheme where the has depends both on the previous token and/or the subsequent token, thereby improving the likelihood that a given token has the watermark applied.  Second, the paper proposes infilling tokens with the greedy approach as a 'prediction' that is then used to construct the hash.  Assuming that the final token sampled in that position is indeed this greedy token, this results in greater detectability.  The authors then empirically evaluate their approach and find it superior to the naive baseline of applying the autoregressive approach whenever a preceding token exists.

**Strengths:**

This paper identifies a clear problem with the well known greenlist approach to watermarking autoregressive language models when applied to diffusion models and proposes a solution that is empirically effective relative to the naive baseline.

**Weaknesses:**

There are several weaknesses with this paper.  First, I think that the several papers including Synth-ID and *GaussMark: A Practical Approach for Structural Watermarking of Language Models* demonstrate that perplexity is a weak proxy of text quality.  Have the authors tested how their approach affects other measures like winrate in AlpacaEval or some other proxy for text quality?

Second, I think that the paper could benefit from further comparison with existing baselines.  There exist watermarking approaches that embed the watermark directly in the weights of a model, with *GaussMark: A Practical Approach for Structural Watermarking of Language Models* being an example; it seems like such an approach could directly translate into diffusion models and potentially be competitive as it does not rely on the autoregressive nature of distribution.  Can the authors comment on this?

Third, I wonder what the increase in sampling time is for this approach over the unwatermarked model?

Fourth, I am a bit confused by the theory behind the predictive approach.  For these kinds of watermarking schemes, many prior works have identified sequence entropy as a key quantity allowing for watermark detectability without a hit in quality, but in higher entropy situations, the predictive approach should be worse.   I understand the ablation that suggests that this is useful in practice, but it would be good for the authors to comment on this?

Fifth, I think that the writing and rigor of the presentation could be improved, e.g., what is the maximum over in equations (7) and (10)?

**Questions:**

See weaknesses.

---

### Official Review · Reviewer_Lmpc · 2025-11-06

**Soundness:** 2
**Presentation:** 3
**Contribution:** 2
**Rating:** 4
**Confidence:** 3

**Summary:**

This paper presents DMark, the first watermarking framework designed specifically for diffusion large language models (dLLMs). Unlike autoregressive models that generate tokens sequentially, dLLMs use iterative denoising with arbitrary token ordering, which breaks existing watermarking methods like KGW. The authors propose three strategies: (1) predictive watermarking that predicts missing context tokens from parallel logit distributions, (2) bidirectional watermarking that exploits both forward and backward dependencies, and (3) predictive-bidirectional watermarking combining both approaches. Experiments on LLaDA and DREAM models show DMark achieves 92.0-99.5\% detection rates at 1\% FPR compared to 49.6-71.2\% for naive KGW adaptation, while maintaining text quality and robustness against various attacks.

**Strengths:**

**Problem Identification:**

- First work addressing watermarking for dLLMs
- Clear articulation of why existing methods fail: out-of-order generation violates sequential dependency assumptions
- Well-motivated problem with practical implications for AI content detection

**Comprehensive Empirical Evaluation:**

- Evaluation across multiple models (LLaDA, DREAM) and datasets (C4, ELI5)
- Extensive parameter sensitivity analysis providing practical deployment guidance
- Robustness testing against various attacks (deletion, insertion, paraphrasing)

**Clear Presentation:** Paper is well-written with effective illustrations (Figure 1) and logical progression from problem to solution.

**Weaknesses:**

**Limited Technical Novelty:**

- *Bidirectional watermarking's* core insight (Eq. 8: define backward green list) is relatively obvious given dLLMs' bidirectional nature. The idea of "if $x_i$ makes $x_{i+1}$ green, then $x_i$ should be in backward green list" is intuitive rather than technically deep
- The combination (predictive-bidirectional) is mechanical—just applying both techniques simultaneously without synergistic interaction
- **No new watermarking primitives:**  The contributions are domain adaptation, not method innovation
- **Missing opportunities for dLLM-specific techniques:** Could exploit iterative refinement (watermark across multiple denoising steps)

**Lack of Theoretical Depth:**

- No formal analysis of detection guarantees under bidirectional constraints
- Missing characterization of prediction error propagation to watermark signal strength
- No study of forward/backward green list correlation and its impact on detection variance

**Experimental Limitations:**

- **Unfair comparison:** Focusing on $w=1$ (single-token context) may disadvantage KGW, which typically uses longer contexts. No analysis of context availability for different window sizes

**Limited Real-World Viability:**

- Paraphrasing attacks reduce TPR to 51% (GPT) / 73% (Dipper)—below practical deployment thresholds
- No discussion of dLLM-specific attack vectors (e.g., exploiting remasking strategies, manipulating confidence scores)

**Questions:**

1. **Novelty justification:** Can you articulate what is technically non-obvious about your approach beyond "adapting KGW to work with bidirectional context"? What insights would not be immediately apparent to researchers familiar with both KGW and dLLMs?

2. **Alternative prediction strategies:** Have you tried sampling $\\hat{x}_{i-1}$ from the logit distribution rather than using $\\arg\\max$? Would this provide better calibration between watermarked and non-watermarked distributions?

3. **Bidirectional independence:** Are forward and backward green list memberships independent? If $x_i \\in G_i(x_{i-1})$, what is $P(x_i \\in G'_i(x_{i+1}))$? This affects detection variance.

4. **Stronger baselines needed:**
   - What if you use oracle ground-truth context when available, predict only when necessary?
   - What if you use bidirectional constraints only when both neighbors naturally exist (no prediction)?

5. **Context window analysis:** You claim longer contexts ($w > 1$) are "rarely available" but provide only one data point (67% for $w=1$). What percentage of positions have $w=2, 3, 5$ available? This is critical for fair comparison.

6. **Theoretical analysis:** Can you provide formal detection guarantees? What is the expected z-score under prediction errors? How does bidirectional bias affect false positive rates?

7. **Computational overhead:** What is the actual runtime overhead of bidirectional watermarking vs. forward-only? Does it negate dLLMs' speed advantage?

8. **Attack surface:** Have you considered adaptive attacks exploiting bidirectional structure? E.g., adversary modifies tokens at positions where both neighbors exist to break both constraints simultaneously?

---

### Note · Authors · 2025-11-27

I have read and agree with the venue's withdrawal policy on behalf of myself and my co-authors.